# Triple-Negative Breast Cancer Histological Subtypes with a Favourable Prognosis

**DOI:** 10.3390/cancers13225694

**Published:** 2021-11-14

**Authors:** Gábor Cserni, Cecily M. Quinn, Maria Pia Foschini, Simonetta Bianchi, Grace Callagy, Ewa Chmielik, Thomas Decker, Falko Fend, Anikó Kovács, Paul J. van Diest, Ian O. Ellis, Emad Rakha, Tibor Tot

**Affiliations:** 1Department of Pathology, University of Szeged, 6725 Szeged, Hungary; 2Department of Pathology, Bács-Kiskun County Teaching Hospital, 6000 Kecskemét, Hungary; 3Department of Histopathology, BreastCheck, Irish National Breast Screening Programme & St. Vincent’s University Hospital, D04 T6F4 Dublin, Ireland; 4School of Medicine, University College Dublin, D04 V1W8 Dublin, Ireland; 5Unit of Anatomic Pathology, Department of Biomedical and Neuromotor Sciences, Bellaria Hospital, University of Bologna, 40139 Bologna, Italy; mariapia.foschini@unibo.it; 6Department of Health Sciences, Division of Pathological Anatomy, University of Florence, 50134 Florence, Italy; simonetta.bianchi@unifi.it; 7Discipline of Pathology, School of Medicine, National University of Ireland Galway, H91 TK33 Galway, Ireland; grace.callagy@nuigalway.ie; 8Tumor Pathology Department, Maria Sklodowska-Curie Memorial National Research Institute of Oncology, Gliwice Branch, 44-102 Gliwice, Poland; Ewa.Chmielik@io.gliwice.pl; 9Department of Surgical Pathology, Dietrich Bonhoeffer Medical Centre, 17036 Neubrandenburg, Germany; thomas.decker@bonhoeffer-klinikum-neubrandenburg.de; 10Reference Centre for Mammography Münster, University Hospital Münster, 48149 Münster, Germany; 11Reference Center for Mammography, 10623 Berlin, Germany; 12Department of Pathology, University of Tübingen, 72076 Tübingen, Germany; Falko.Fend@med.uni-tuebingen.de; 13Department of Clinical Pathology, Sahlgrenska University Hospital, 41 345 Gothenburg, Sweden; aniko.kovacs@vgregion.se; 14Department of Pathology, University Medical Centre Utrecht, 3584 CX Utrecht, The Netherlands; P.J.vanDiest@umcutrecht.nl; 15Department of Histopathology, University of Nottingham and The Nottingham University Hospitals NHS Trust, Nottingham City Hospital, Nottingham NG5 1PB, UK; ian.ellis@nottingham.ac.uk (I.O.E.); Emad.Rakha@nottingham.ac.uk (E.R.); 16Pathology & Cytology Dalarna, Falun County Hospital, 791 82 Falun, Sweden; tibor.tot56@gmail.com

**Keywords:** acinic cell carcinoma, adenoid cystic carcinoma, fibromatosis like metaplastic carcinoma, low-grade adenosquamous carcinoma, mucoepidermoid carcinoma, secretory carcinoma, tall cell carcinoma with reversed polarity, triple negative breast cancer

## Abstract

**Simple Summary:**

Breast cancers that lack expression of the predictive markers oestrogen receptor, progesterone receptor and human epidermal growth factor receptor-2 are known as triple-negative breast cancers (TNBCs) and are generally considered to have a poor prognosis. As available targeted treatments are not effective, aggressive chemotherapy is frequently advocated for patients with TNBC. It is now becoming apparent that TNBC is not one entity but constitutes a range of malignancies with different clinical behaviour. This paper reviews 7 distinct histological subtypes of TNBC where the overall prognosis is favourable, and aggressive systemic treatment is generally not indicated. Their recognition and separation from the larger group of no special type TNBC are important. The members of the European Working Group for Breast Screening Pathology review the morphology, known molecular features and reported outcomes, and formulate a consensus statement regarding the approach to the subtypes that are associated with a favourable prognosis.

**Abstract:**

Triple-negative breast cancers (TNBC), as a group of tumours, have a worse prognosis than stage-matched non-TNBC and lack the benefits of routinely available targeted therapy. However, TNBC is a heterogeneous group of neoplasms, which includes some special type carcinomas with a relatively indolent course. This review on behalf of the European Working Group for Breast Screening Pathology reviews the literature on the special histological types of BC that are reported to have a triple negative phenotype and indolent behaviour. These include adenoid cystic carcinoma of classical type, low-grade adenosquamous carcinoma, fibromatosis-like metaplastic carcinoma, low-grade mucoepidermoid carcinoma, secretory carcinoma, acinic cell carcinoma, and tall cell carcinoma with reversed polarity. The pathological and known molecular features as well as clinical data including treatment and prognosis of these special TNBC subtypes are summarised and it is concluded that many patients with these rare TNBC pure subtypes are unlikely to benefit from systemic chemotherapy. A consensus statement of the working group relating to the multidisciplinary approach and treatment of these rare tumour types concludes the review.

## 1. Introduction

Our understanding of the biology of human breast cancer (BC) has evolved exponentially since the first documented written reference to BC in the Edwin Smith papyrus in 3000 B.C. and ancient Greek times when Hippocrates perceived the disease as a single mortal one [1]. BC is now a collective term for a heterogeneous group of diseases, variously classified according to the putative cell of origin, histological type, tumour grade, molecular markers, tumour stage and other clinical and biological variables that correlate with outcome and indicate potential response to specific therapies.

Since the seminal work of Perou and colleagues, BC has been classified into molecular (intrinsic) subtypes according to their gene expression profiles (GEPs) [2]. The different molecular subtypes of BC are associated with different prognoses [3,4] with luminal A tumours having the best outcome and basal-like carcinomas demonstrating the worst survival [3,4]. In daily practice immunohistochemistry (IHC) is used as a surrogate classification system to categorise BCs, according to oestrogen receptor (ER), progesterone receptor (PR) and human epidermal receptor growth factor 2 (HER2) expression, into luminal (ER+ and/or PR+), HER2 positive and triple-negative tumours. Whilst there is considerable overlap between basal-like tumours and triple-negative breast cancers (TNBCs), these tumours are not identical [5] and the surrogate classification has a number of mismatches when compared to the GEP based classification [6]. As a group, TNBCs have worse overall survival (OS) than stage-matched non-TNBCs. An analysis of the Survival, Epidemiology and End Results database with 18,855 TNBCs and 139,503 non-TNBCs grouped according to their hormone receptor (HR) and HER2 statuses into HR+HER2-, HR+HER2+ and HR-HER2+, showed that the OS of TNBCs was significantly worse for all substages and groups of non-TNBCs with hazard ratios between 1.40 and 4.19 (except for substages IA and IB of HR-HER2+ cancers, where the hazard ratios of 1.21 and 1.76 failed to be significant) [7].

It is now recognized that TNBC constitutes a heterogeneous group of tumours at the molecular level [8] with six subgroups originally identified [9,10], later refined to four: basal-like 1 and 2 tumours, that differ in their immune-response, mesenchymal and luminal androgen receptor tumours, recognised in the 5th edition of the World Health Organisation (WHO) Classification of Breast Tumours [11]. At present, no clinically verified molecular assay exists for the optimal classification of TNBC [11] and the prognostic and predictive relevance of this classification of TNBC has yet to be defined.

At the morphological level, TNBC is also a heterogeneous disease with high- and low-grade variants and a corresponding spectrum of biological behaviour. However, due to its frequent association with high histological grade and aggressive biological behaviour, the management of TNBC is a major focus in the field of medical oncology due to the lack of options for targeted therapy [12]. Although the prognosis of non-metastatic TNBC is influenced by a number of variables besides the triple negative phenotype [13,14,15], it is not an unexceptional practice to recommend adjuvant systemic chemotherapy (CT) to (nearly) all patients with TNBC to improve survival.

This review, on behalf of the European Working Group for Breast Screening Pathology (EWGBSP, for details, see: ewgbsp.org) aims to highlight the pathology and clinical behaviour of some relatively rare TNBC subtypes, recognized by the WHO classification [11], that appear to be associated with a favourable prognosis (Figure 1). Recognition of these TNBC histological subtypes is important to avoid a generalised approach to the systemic management of patients with TNBC. The review focuses on the pure variants of these tumours as prognosis may be modified by other components in mixed tumours. Some recognised TNBC subtypes with unique morphology, biology or behaviour, e.g., metaplastic spindle cell, matrix producing, squamous and apocrine carcinoma, are not discussed as current evidence does not support changing adjuvant treatment approaches compared to TNBC of no special type (NST). The tumours discussed below are listed in alphabetical order with emphasis on the key clinical and morphological features and biological behaviour.

## 2. Acinic Cell Carcinoma (ACC) of the Breast

### 2.1. Definition, Main Features, Diagnostic Clues and Differential Diagnosis

Acinic cell carcinoma (ACC) of the breast is a subtype of TNBC, morphologically similar to ACC of the salivary glands (Figure 1A). It mainly affects adult women, presenting as a nodule/mass ranging in size from 10 to 52 mm (average 28 mm; summary in Appendix A) [16,17,18,19,20,21,22,23,24,25,26,27,28,29,30,31,32,33,34,35,36,37].

ACC can show a wide spectrum of architectural patterns. At one end of the spectrum, ACC is composed of a proliferation of small round glands (microglandular pattern), lined by a single layer of cuboidal to columnar epithelial cells resembling the acinar cell structure of salivary glands and microglandular adenosis of the breast. Eosinophilic/amphophilic secretions may be present in the glandular lumina. An in situ component may be associated. Tumour cells are polygonal, sometimes with clear (hypernephroid), PAS-positive, finely granular eosinophilic or basophilic cytoplasm and prominent nucleoli after diastase digestion. The eosinophilic zymogen-type cytoplasmic granules may be large and coarse, resembling intestinal Paneth cells. Mitoses are variably present [11,38]. Tumour cells show serous differentiation verified by a positive immunoreaction for amylase, lysozyme and α-1 antichymotrypsin. Epithelial membrane antigen (EMA), S-100, low molecular weight cytokeratins (CKs) are usually positive; and gross cystic disease fluid protein 15 (GCDFP-15) may be positive. No pathognomonic genetic alterations have been identified in three breast ACCs analysed by whole-exome sequencing. The alterations identified were similar to those seen in conventional TNBC of NST [39]. On the basis of the small series analysed, commonly occurring mutations affect the following genes: *PIK3CA, KMT2D, ERBB4/ERBB3, NEB, BRCA1, MTOR, CTNNB1, INPP4B* and *FGFR2* [11,34,40].

At the other end of the spectrum, ACC is composed of circumscribed solid nests of variable size with comedo-type necrosis, prominent nuclear atypia and increased mitotic activity. ACC may also be admixed with NST or metaplastic elements. ACC may show intra-tumour heterogeneity with both well-differentiated microglandular and less-differentiated solid areas. Thorough sampling of ACC is important to identify components that are likely to change the perceived risk of these tumours and may explain the clinical events reported in some studies (see below).

### 2.2. Clinical Correlations, Treatment and Outcome Data

ACC was originally described in 1996 [16], with a limited number of reported cases in the English literature. Foschini et al. [41] listed 45 ACCs with a further 2 cases since reported [36,37]. Some studies include very limited or no follow-up data. Appendix A is an adapted version of the table reported by Foschini et al. [41] including only those patients for whom follow-up data were available (follow-up range 3 to 184.8 months, average 73 months) and updated with recent literature. In approximately half of the patients (16/35) follow-up was less than 2 years (Appendix A).

At presentation most patients with ACC regardless of its subtype (24/35, 68.6%) had pT2 tumours. Axillary lymph node metastases were detected in 8/31 (26%), only one of which was associated with a pT1c tumour. In most patients, regional axillary metastases were limited to 1 or 2 lymph nodes. One patient had a high axillary metastatic burden [34] and died of disease 24 months after presentation [34]. Local recurrences were observed in 3 patients [18,34,35], one of whom had a large tumour treated with local excision only [18] and one who developed recurrence despite aggressive treatment [35]. No data on surgical treatment were reported for the remaining patients [34]. Distant metastases (lung, bone, liver) are reported in 3 patients, two of whom died from their disease [20,23,25]. Sixteen patients (comprising all with distant metastases) were treated with adjuvant therapies (Appendix A). ACC, when associated with other types of high-grade conventional BC, can show aggressive behaviour mimicking that of the higher grade component. Sardana et al. [42] reported a patient with ACC, associated with NST and metaplastic BC components who developed meningeal metastases and died from her disease.

It is difficult to draw firm conclusions regarding the prognosis of breast ACC from available literature in view of morphological variation within and between tumours (pure and mixed types) in the published studies. It is our opinion that low-grade pure ACC is a bland tumour type that overlaps with microglandular adenosis and is associated with indolent biological behaviour, and is, therefore, unlikely to benefit from aggressive adjuvant chemotherapy. ACCs with high-grade areas or admixed with other BC types are likely to behave more aggressively and may account for some of the reported events in the literature. Currently, there is no evidence to support withholding systemic chemotherapy in ACC with high-grade features if clinically indicated. Further data on their response to therapy is needed.

## 3. Classic Adenoid Cystic Carcinoma (CAdCC)

### 3.1. Definition, Main Features, Diagnostic Clues and Differential Diagnosis

Adenoid cystic carcinoma (AdCC) is a rare salivary gland type tumour in the breast, first described as a cylindroma of the breast by Billroth in 1856 [43,44]. According to the current WHO classification of breast tumours three subtypes of AdCC are recognized: classic, solid basaloid, and AdCC with high-grade transformation [11].

Classic AdCC (CAdCC) is usually unifocal [44] but may be multifocal [45]. Approximately half of all cases arise in the subareolar region [46]. Microscopically, the tumour is composed of epithelial (luminal) and myoepithelial (abluminal) neoplastic cells arranged in tubular and cribriform patterns with intervening lumina containing mucin and pseudolumina containing reduplicated basement membrane material (Figure 1B) [11]. A minor solid growth pattern is occasionally observed in the classical subtype but is more common in the solid basaloid variant and AdCC with high grade transformation. A solid growth pattern and high-grade features are histological signs of likely aggressive biological behaviour in AdCC.

The cells of the epithelial component are positive for CK7, CK5/6, CK 8/18 and CD117 [47]. The myoepithelial/abluminal cells express p63, smooth muscle actin and basal CKs: CK5/6, CK14, CK17. CAdCC generally displays a triple-negative immunophenotype, although ER-positive cases also occur (Appendix A) [46,48,49,50,51,52,53,54,55,56,57,58,59,60,61,62,63,64,65,66,67,68]. CAdCC frequently expresses EGFR, an immunohistochemical marker of the “basal-like” phenotype, without gene amplification [69]. CdACC may also express a truncated form of the ER receptor-alpha, which is not detected by antibodies in general use [70] and the significance of which is not known. CAdCC has a low proliferative fraction [11], in keeping with the overall good prognosis associated with this tumour.

Most AdCCs investigated to date have harboured the *MYB-NFIB* fusion gene. AdCCs lacking the *MYB*-*NFIB* fusion gene may show *MYBL1* rearrangements or *MYB* amplification [71]. The most frequently mutated genes in AdCC include *MYB*, *BRAF*, *FBXW7*, *SMARCA5*, *SF3B1*, and *FGFR2*. AdCCs appear to lack somatic mutations in the *TP53*, *PIK3CA*, *RB1*, *BRCA1* and *BRCA2* genes which are often mutated in TNBC, NST and basal-like breast cancers [72].

CAdCC shows morphological overlap with a variety of benign, atypical and good-prognosis malignant breast lesions including collagenous spherulosis, syringomatous adenoma, adenomyoepithelioma, cribriform ductal carcinoma in situ (DCIS) and invasive cribriform carcinoma (ICC) [47]. CD117, which is positive in the luminal cells of CAdCC, is negative in collagenous spherulosis. Calponin and smooth muscle myosin heavy chain, negative in abluminal CAdCC cells, are strongly positive in collagenous spherulosis [73]. Syringomatous adenoma lacks cytologic atypia [47]. The myoepithelial cells in adenomyoepithelioma are positive for calponin while the abluminal cells in CAdCC are negative [47]. Cribriform DCIS shows regular contours with peripheral myoepithelial cells but no internal reactivity for myoepithelial cell markers [47]. The stroma of ICC tends to be desmoplastic, whereas in CAdCC the stroma shows myxoid change around the cribriform islands. ICCs are typically ER-positive, with 69% also PR positive, and HER2 negative [11].

### 3.2. Clinical Correlations, Treatment and Outcome Data

A review of the published literature identified database studies and small case series in addition to single case reports concentrating on tumours associated with an unfavourable outcome (Appendix A) [46,49,50,51,52,53,54,55,56,57,58,59,60,61,62,63,64,65,66,67,68]. The reported series often included a mixture of AdCC tumours with CAdCC rarely studied in isolation. This may be related to the fact, that prior to the 2019 edition, the WHO classification of breast tumours did not categorically distinguish between the three morphological variants of AdCC, although the solid form had been described in the 4th edition [74].

Taking account of the publications to date, the median age of patients with AdCC ranged from 58 to 66 years. Data on the ethnical differences are limited, but larger registry-based series suggest that approximately 85% of patients diagnosed in the USA were whites [46,48,50,52,53]. The majority of patients presented with a palpable mass (85.7%) and underwent lumpectomy, breast-conserving surgery (BCS) or mastectomy. Lymph node status was assessed in all reviewed cases, and nodal involvement ranged from 0 to 15% with the exception of a small series (*n* = 19), where it reached 27% [60]. Although grading was applied to 57% of the collected cases in Appendix A, the methods were different and included the original Bloom and Richardson system, the Nottingham (modified Bloom and Richardson) system, nuclear grade (NG1-3), a three-tiered grading for salivary gland AdCC (G1-G3), and low grade vs. high grade.

Adjuvant radiotherapy was administered to 17–66% of the patients, all of whom had BCS. The percentage of patients who received chemotherapy ranged from 4 to 66%, not exceeding 25% in most studies (Appendix A).

The literature on rare entities including CAdCC is limited and of questionable value without a detailed analysis of pathological characteristics. Slodkowska et al. presented a precise pathological approach with predictors of clinical outcome [68] and confirmed that pure CAdCC has an excellent prognosis. In contrast, solid-basaloid AdCC is an aggressive variant as supported by survival analysis [68]. Slodkowska et al. also confirmed that the Nottingham grading system was a strong predictor of the behaviour of breast AdCC. Perineural invasion was more often seen in tumours that recurred (19% overall and 50% in recurring cases). Neovascularisation also emerged as a new independent predictor of aggressive behaviour in this disease [68].

Therefore, CAdCC in its pure form, without solid or transformed components has a good prognosis and patients with this entity are unlikely to derive benefit from adjuvant chemotherapy.

## 4. Fibromatosis-l ike Metaplastic Carcinoma (FLMC)

### 4.1. Definition, Main Features, Diagnostic Clues and Differential Diagnosis

Fibromatosis-like metaplastic carcinoma (FLMC) represents a rare low-grade subtype of metaplastic breast carcinoma (MBC). With a total of 70 cases reported in the peer-reviewed English language literature (Appendix A) [75,76,77,78,79,80,81,82,83,84,85,86,87,88], FLMC is likely to account for significantly less than 1% of all MBCs. Due to its bland morphology, FLMC is diagnostically challenging and is likely to account for a higher relative percentage of tumours in referral centres, diagnosed at the time of primary presentation or on review following a recurrence of an incompletely excised FLMC originally (mis)diagnosed as a benign entity.

FLMC is characterized by a proliferation of spindled fibroblast-like cells and stellate myofibroblast-like cells, which compose more than 95% of the total tumour area, and histologically resembles fibromatosis. The neoplastic cells are cytologically bland with absent to minimal nuclear atypia and pale eosinophilic cytoplasm. Within the same tumour, there may be a gradual transition from cells with thin, slender, spindled nuclei with tapered ends to cells with plump, round to oval nuclei with discrete nucleoli within finely distributed chromatin. High nuclear grade is not seen [11,75,76]. Mitotic figures are either completely absent or rare [75,76]. Necrosis is not identified. Neoplastic squamous or glandular epithelial elements may be admixed with the spindle cells but should account for less than 5% of the total tumour area and FLMC should not contain any other differentiated component (Figure 1C) [11]. The epithelial nature of the spindle cells is often only recognisable on immunohistochemistry and may be focal. The use of a panel of CKs is advisable to confirm the diagnosis, including broad-spectrum, high and low molecular weight CKs e.g., CK AE1/AE3, MNF116, 34betaE12, CK8/18, CK5/6, and CK14. Most tumours also express p63. Expression of vimentin and other myoepithelial markers is variable. ER, PR and HER2 are negative [75,76,89,90,91].

FLMC and low-grade adenosquamous carcinoma (LGASC) may represent a spectrum, composed of similar histological components (spindle, glandular and squamous), with the glandular and squamous components accounting for less than 5% in FLMC and a significantly greater percentage of LGASC. It is our experience that most of the missed LGASC lack glandular or squamous components and just feature bland-looking spindle cell proliferation that express epithelial markers. Both FLMC and LGASC may incorporate sclerosing papillary or adenomyepitheliomatous foci with which they share molecular alterations including mutations in the *PIK3CA* and *SF3B1* genes, suggesting a possible pathogenetic link.

In clinical practice pathologists often differentiate FLMC from fibromatosis using IHC markers. Fibromatosis lacks CK expression and usually shows abnormal nuclear localization of beta-catenin. The two entities are also different at the molecular level, with exon 3 *CTNNB1* mutations being highly specific for desmoid type fibromatosis among spindle cell lesions of the breast, and *APC* mutations also being relatively common in fibromatosis [92]. FLMC should also be distinguished from some more aggressive malignant lesions with overlapping morphology. Spindle cell MBC is associated with recognisable nuclear atypia and is usually larger. Pure FLMC should not display intermediate/high-grade nuclear features or be admixed with other BC types such as NST or lobular carcinoma. The presence of one or more of the latter components in a tumour with low-grade spindle cell morphology excludes a diagnosis of FLMC [11,89].

### 4.2. Clinical Correlations, Treatment and Outcome Data

Since the first detailed description of the histological criteria for the diagnosis of FLMC by Gobbi et al. in 1999 (adopted by the current WHO classification), a total of 70 cases have been published in the English-language literature with follow-up data available for 41. Overall, FLMC shows clinically indolent behaviour with a high tendency for local recurrence (14/41 cases), but with a low potential for regional lymph node (3/41 cases) or distant metastases (5/41 cases) (Appendix A) [75,76,77,78,79,80,81,82,83,84,85,86,87,88].

Our review of the literature suggests that the risk of local recurrence is at least partly related to inadequate local resection [75,76,80]. Lack or incomplete excision of these tumours may lead to progressive growth and acquisition of additional mutations and more aggressive behaviour. It is our view that the metastatic events reported in the literature for cases of FLMC likely reflect the inclusion of tumours with a high-grade spindle cell component (MBC). The association between tumour diameter and the risk of distant metastasis assumed by individual authors is not supported by data in the published literature (Appendix A).

There are no evidence-based treatment guidelines for FLMC. Regarding local therapy, wide excision appears to prevent local recurrence [75,76,80]. The available data do not support the benefit of adjuvant radiotherapy or chemotherapy in reducing the risk of local recurrence or metastasis (Appendix A) [75,76,77,78,79,80,81,82,83,84,85,86,87,88]. However, some authors, including the present authors, advocate the use of adjuvant radiotherapy for more bulky lesions [90]. Because of the very low risk of regional lymph node metastases, several authors argue against axillary lymph node dissection in pure FLMC particularly in the management of small tumours [75,89].

## 5. Low-Grade Adenosquamous Carcinoma (LGASC)

### 5.1. Definition, Main Features, Diagnostic Clues and Differential Diagnosis

Low-grade adenosquamous carcinoma (LGASC) of the breast is a rare variant of MBC with a favourable outcome. Histologically, these tumours are composed of well-developed glandular and tubular formations intimately admixed with solid nests of squamous cells arranged in a haphazard, infiltrative pattern in a spindle cell background, typically associated with a peripheral lymphocytic infiltrate (Figure 1D) [11]. Cytological atypia is mild and proliferative activity, measured by mitotic activity and Ki-67 index, is low [93].

LGASC constitutes a distinct genetic entity among MBCs, characterised by high rates of *PIK3CA* mutations, lack of *TP53* mutations, a triple-negative phenotype and lack of androgen receptor (AR) expression [93]. There is generally expression of luminal (CK7, CK8) and basal (CK5, CK14) CKs and squamous (myoepithelial) markers p63 and p40.

LGASC should be distinguished from other TNBC and basal-like carcinomas that are associated with aggressive biological behaviour, i.e., high-grade MBCs that may have areas of squamous metaplasia and spindle cells. Nuclear pleomorphism, mitoses, necrosis, a prominent malignant appearing sarcomatous spindle cell component and solid nests of atypical squamous cells support classification as spindle cell MBC and/or squamous carcinoma rather than as LGASC.

### 5.2. Clinical Correlations, Treatment and Outcome Data

Rosen and Ernsberger [94] described this entity in 1987 emphasising that, despite the presence of metaplastic elements, this tumour displays low-grade histological features. In keeping with their low-grade morphological features, the majority of LGASCs have an excellent prognosis with a low incidence of lymph node metastases.

A review of the published English literature identified all reports of LGASC of the breast. Case reports/series without data on treatment and outcome were excluded. A total of 15 publications dating from 1987 to the present describing 92 cases of LGASC with data on treatment and outcome were identified (Appendix A) [93,94,95,96,97,98,99,100,101,102,103,104,105,106,107]. Patient age ranged from 19 to 88 years. The tumours were unilateral apart from one patient with bilateral LGASC. The majority of patients presented with a palpable mass (83%). Patients underwent BCS or excisional biopsy (67%) or mastectomy (33%). Lymph node status was assessed in 34 patients (37%) with only one incidence of nodal metastasis reported. Adjuvant radiotherapy was administered to 18 patients, all of whom had BCS or excisional biopsy, and seven patients received chemotherapy.

Early studies [94,107] reported relatively high rates of local recurrence, in 36% and 20% of patients respectively all of whom had been treated initially by excisional biopsy only. In more recent literature, only a single patient had subsequent progression of the disease [101].

Although the literature on this rare tumour is limited, the available data strongly suggest that this is a malignant tumour with an indolent course and a favourable prognosis. Recent studies demonstrate a low rate of local recurrence with no nodal or distant metastases, anyway the definitive and optimal treatment for LGASC has yet to be determined. The benefit of adjuvant radiotherapy has not been studied yet. Chemotherapy does not appear to be warranted and adjuvant hormonal treatment is not indicated because of the triple negative phenotype of these tumours. At present, the clinical management of LGASC requires complete surgical excision followed by adjuvant radiotherapy in case of conservative surgery.

## 6. Mucoepidermoid Carcinoma (MEC) of the Breast

### 6.1. Definition, Main Features, Diagnostic Clues and Differential Diagnosis

Mucoepidermoid carcinoma (MEC) of the breast is an invasive carcinoma composed of mucoid, epidermoid and intermediate cells analogous to the tumour of the same name encountered in salivary glands (Figure 1E) [11]. In contrast to minor salivary glands, where this entity is the most common malignancy, breast MEC is rare, with less than 50 cases reported to date in the English literature.

Breast MEC affects middle-aged and elderly women. It presents as a unilateral nodule, sometimes with a cystic component. On ultrasound examination, MEC can simulate a benign lesion with enhancement rather than shadowing [108].

Morphologically, it is similar to salivary gland MEC with a great variety of patterns. Histological grade ranges from low to high and grading can be performed by applying the systems for both the breast and the salivary gland [109].

Low-grade MEC is more frequently cystic, composed of mucoid, epidermoid and basaloid cells. Mucoid cells usually line the cystic spaces and may be intermingled with columnar cells devoid of intracellular mucous. The epidermoid cells have eosinophilic or clear cytoplasm. True keratinization with squamous pearls is not seen and, when present, should direct the diagnosis toward adenosquamous carcinoma rather than MEC. Basaloid cells are usually located at the periphery of the neoplastic cysts and nests. High-grade MEC is more frequently solid and is composed of the same cell types as low-grade MEC, but with a higher degree of nuclear atypia and a higher mitotic count. Necrosis may be present. Rare cases of intermediate grade MEC have been reported [41,109].

Immunohistochemistry assists the diagnosis as each cell type has a characteristic profile. Specifically: mucoid cells are positive for low molecular weight CKs, e.g., CK7, while high molecular weight CKs, e.g., CK14, CK5, and p63 stain the epidermoid and basaloid cells. All reported breast MECs to date have shown a triple negative phenotype.

A few cases have been studied with molecular analyses, demonstrating *CRTC1-MAML2* rearrangement, similar to that observed in the salivary gland counterpart [110,111,112,113].

### 6.2. Clinical Correlations, Treatment and Outcome Data

Histological grading is very important for prognostic purposes [114]. In a recent review, Ye et al. identified 42 cases of breast MEC published from 1979 when Patchefsky et al. first described this tumour type in the breast [115], of which 19 were classified as low grade, 3 as intermediate grade, 17 as high and 3 had no grade reported. Their table has been completed with treatment information and used to summarize the low and intermediate grade cases (Appendix A) [109,111,115,116,117,118,119,120,121,122,123]. In the non-high grade MEC group, 7 patients had BCS (6 low grade and 1 with intermediate grade) and 16 had mastectomy (13 low grade and 3 intermediate grade) [108]. Axillary nodal status was reported in 11 patients with low-grade MEC, with lymph node metastases (3/18) in only one (9.1%; 95%CI (confidence interval): 0.5–42.9%) [118]. Of the 2 patients with intermediate MEC and known nodal status, one had a positive sentinel lymph node [111]. Follow-up data were available in 19 patients with low-grade MEC, ranging from 3 to 156 (median 48) months, and in two patients with intermediate MEC with 3- and 8-months follow-up, respectively. At present, none of the patients with low or intermediate grade MEC has developed metastases or died. There is one report of a patient with low grade breast MEC who developed a high grade MEC recurrence [122] but was alive and well at 156-month follow-up. In contrast, distant metastases and progression to death occurred in 4 patients with high-grade MEC [108].

According to the data reported, low-grade MEC patients have a good overall prognosis, even though adjuvant chemotherapy was not administered to some of them. The review of the data, therefore, supports that adjuvant chemotherapy is not indicated in these patients simply on the basis of the triple-negative phenotype.

## 7. Secretory Carcinoma (SC)

### 7.1. Definition, Main Features, Diagnostic Clues and Differential Diagnosis

Secretory carcinoma (SC) is a rare tumour accounting for less than 0.02% of all BCs [11,124]. The name “secretory” was assigned due to the presence of eosinophilic, intra- and extracellular secretory material. The first case may have been described by Levings in 1917, but the first series and its recognition as a childhood cancer (hence the original name “juvenile breast carcinoma”) was by McDivitt and Stewart [125]. As the tumour is not restricted to the paediatric population and more commonly occurs in adults, Tavassoli and Norris re-named this tumour ‘secretory carcinoma’ in 1980 [126]. The average age at presentation is 53 years (3–87 years). It mainly occurs in females [11] but has also been reported in males, summarized by Ghilli et al. [127], in whom it may exhibit a more aggressive clinical course [128]. A similar tumour type may be seen in salivary glands (previously referred to as mammary analogue secretory carcinoma, MASC) and occasionally in the thyroid gland, skin and respiratory tract [129]. Rare cases of secretory carcinoma in situ have also been reported [130,131].

Grossly, the typical appearance is that of a rounded, circumscribed, greyish-white mass, sometimes with tan to yellow discolouration [11]. This appearance may mimic benign lesions, e.g., fibroadenoma, on imaging.

The tumour cells have vacuolated cytoplasm with abundant intracellular material that is Periodic Acid Schiff (PAS), PAS diastase or alcian blue positive. Extracellular secretory material is also seen. The nuclei of the tumour cells are small and bland. Mitotic activity is low. The architectural arrangement varies with solid, ductal/tubular, microcystic (“honeycomb”, mimicking thyroid follicles) and mixed patterns (Figure 1F). A case with a predominant papillary pattern has also been reported [132].

Immunohistochemically, most SCs are triple negative and may express basal markers (e.g., cytokeratin 5/6 and EGFR). Weak ER and PR positivity are observed in some tumours. SCs consistently exhibit positivity for S-100 and α-lactalbumin [11,133].

SC is characterised by a recurrent chromosomal rearrangement, t(12:15)(p13:q25), that is detected in 75–92% of tumours. This leads to fusion between E26 transformation specific translocation variant 6 (*ETV6*) and neurotrophic receptor tyrosine kinase 3 (*NTRK3*) [134] with potential for targeted therapy using larotrectinib and entrectinib. This fusion may also be detected in congenital fibrosarcoma, cellular mesoblastic nephroma, acute myeloid leukaemia, ALK-negative inflammatory myofibroblastic tumour and radiation-induced papillary thyroid carcinoma [130,135] but appears to be specific for SC in the breast context. Harrison et al. suggested the use of pan-TRK (tropomyosin receptor kinase) IHC for all TNBCs to identify *NTRK* rearrangements [136] but the specificity of this approach is questionable [137]. Additional mutations like *TERT* promoter mutations and loss of *CDKN2A*/B have also been reported and may be associated with an aggressive course in SC but further studies are needed to identify predictive markers of a more aggressive clinical course [130].

The differential diagnosis of SC includes invasive apocrine carcinoma, ACC, juvenile papillomatosis with apocrine metaplasia, microglandular adenosis and other breast lesions with secretory type features including cystic hypersecretory hyperplasia, cystic hypersecretory in situ carcinoma, collagenous spherulosis, microglandular adenosis and lactational type change. Careful evaluation of morphology and judicious use of immunohistochemistry greatly assists accurate diagnosis. Apocrine carcinomas can generally be distinguished by their characteristic androgen receptor expression. Microglandular adenosis also expresses S100 but has specific morphological features with a typically infiltrative pattern.

### 7.2. Clinical Correlations, Treatment and Outcome Data

Since the description of the initial series by McDivitt and Stewart, several hundred SCs have been reported, mostly as single case reports, small single institutional series and two larger database analyses [138,139]. These are summarized in Appendix A [124,125,127,129,130,131,132,138,139,140,141,142,143,144,145,146,147,148]. In 2018, Garlick reviewed 89 cases previously published and these are included in the table on the basis of his review [144]. The largest series, from the National Cancer Database, includes 246 SCs [139]. Data on racial differences are limited, but one series suggested a slightly greater incidence among African American women in the USA [139]. It is noteworthy that 64% of the tumours with known receptor status were ER-positive. Although this may cast doubt on the appropriateness of histological typing in some cases, it is consistent with the observation that, while most SCs are triple negative, it is not uncommon to see weak ER and PR expression [11]. Several examples of SCs with a proven diagnostic translocation present have demonstrated ER positivity.

The cases and series listed in Appendix A represent only those reported in the English language literature. About half of the patients with SC were treated with mastectomy, including simple, modified radical and radical, the latter reflecting accepted practice at the time this tumour was first described. Lymph node involvement is influenced by tumour size and lymphovascular invasion, and ranged between 0 and 50% in reports, with a rate of 32% in the National Cancer Database series of 246 patients [138] and 15–30% in the series reported by Altundag et al. [148].

Patients treated with BCS were likely to receive adjuvant radiotherapy, although some did not [144].

Fifty-five years ago, when the first series was published, adjuvant therapy was not widely administered. Of seven patients treated with surgery alone and a median follow-up of 10 years, only one developed two consecutive local recurrences leading to mastectomy and had an uneventful course after two years of follow-up. More recent case reports and series have documented the use of adjuvant chemotherapy, but more than half of the cases summarized in Appendix A did not receive systemic treatment. De-escalating systemic treatment has also been advocated [143]. Rare tumours may demonstrate high-grade transformation similar to translocation driven salivary gland-type carcinomas [129] and may benefit from targeted anti-TRK treatment [141,149].

In general, the follow-up data suggest that SC generally pursues an indolent clinical course, even in patients with lymph node-positive disease [128,130], with 5- and 10-year disease-free survival rates of over 90% [136]. Breast cancer-specific survival at 5 and 10 years were 94% and 91%, respectively, in the Survival, Epidemiology and End Results (SEER) analysis of 83 patients [138]. The data, therefore, support that SC does not require adjuvant systemic therapy simply because of the TNBC phenotype.

## 8. Tall Cell Carcinoma with Reversed Polarity (TCCRP)

### 8.1. Definition, Main Features, Diagnostic Clues and Differential Diagnosis

Tall cell carcinoma with reversed polarity (TCCRP) is a rare special type of BC. It is characterized by tall columnar cells arranged in nests with a predominant solid papillary pattern and demonstrates reversed nuclear polarity, i.e., the nuclei are located at the apical rather than at the basal aspect of the cells [11]. The absence of myoepithelial cells at the periphery of the tumour nests is also an essential diagnostic criterion. An in situ component, as reported in some papers [150,151,152], does not exclude the diagnosis.

The histological appearances may vary with both structural (papillae, dense eosinophilic colloid-like material in follicle-like structures and occasional calcifications) and nuclear (groves, pseudo-inclusions, tall columnar cells) similarities to the tall cell variant of thyroid papillary carcinoma observed; the first name of this entity was based on this resemblance (Figure 1G) [150]. The neoplastic cells have eosinophilic and granular cytoplasm, rich in mitochondria. The stroma between the tumour cell nests is generally dense, and as in most solid papillary carcinomas, the nests are often surrounded by a delicate rim of capillaries. Basement membrane and smooth muscle stains on immunohistochemistry may mimic a myoepithelial cell layer or an epithelium-bound basement membrane. In keeping with a low-grade appearance, mitotic figures are rare and the Ki67 proliferation index is low.

Most TCCRPs are triple negative although some may demonstrate weak hormone receptor positivity [11], a phenotype that more closely resembles hormone receptor negativity than positivity [153]. TCCRP may also express androgen receptors. Co-expression of high- and low-molecular-weight CKs is listed as a desirable diagnostic criterion in the WHO classification [11]. However, expression of CK5 may show a mosaic-like pattern and lead to a mistaken diagnosis of a benign, hyperplastic proliferation [154]. Unlike typical hyperplastic proliferations, however, CK14 is not expressed in TCCRP [152]. Neuroendocrine and thyroid markers are negative in these lesions. Breast markers are expressed in TCCRP, and breast origin may be proven by the use of multiple markers e.g., GATA-3, GCDFP-15 and mammaglobin. The immunostaining with anti-mitochondrial antibody is strongly positive, especially at the basal aspect of the neoplastic cells, evidencing the reverse polarization [155].

The most characteristic molecular alteration in TCCRP is the presence of a p.R172 hotspot mutation in the *IDH2* gene [156,157,158,159,160] which is very uncommon in other breast tumours. A p.R120 mutation of the *IDH2* gene has also been described [161]. *PIK3CA* mutations, common in many types of BC, are also frequent in TCCRP [157,159]. PRUNE2 mutations and *ATM* mutations have also been described in 6/9 and 2/3 cases, respectively [154,158]. In keeping with the IHC profile excluding a thyroid origin, *RET/PTC* rearrangements and *BRAF* mutations have not been reported in TCCRP.

Differential diagnostic problems may arise especially in core needle biopsies, when the full architecture cannot be evaluated.

The main differential diagnosis of TCCRP is metastatic thyroid papillary carcinoma, with which the morphology overlaps [162]. The immunohistochemistry and molecular features described above, including breast marker positivity, exclude this entity. As a triple-negative tumour of the breast, it must be separated from other triple-negative/basal-like carcinomas that are associated with aggressive behaviour, i.e., metaplastic or basal-like carcinomas which also express CK5 or CK5/6. The typical low grade and thyroid-like nuclear features, in addition to the reversed nuclear polarity, assist this distinction.

The thin vascular channels around the tumour cell nests, when highlighted by basement membrane or smooth muscle immunostains, may mimic myoepithelium and therefore an in situ or benign process [155]. The use of IHC markers that do not label vessels (e.g., p63, CD10) may help to exclude the presence of myoepithelial cells.

The low-grade nuclear pattern, the papillary growth and the CK5 staining pattern may point to a usual type hyperplastic proliferation [154] which may be excluded by the absence of CK14 staining and lack of a mosaic pattern of ER and PR expression.

### 8.2. Clinical Correlations, Treatment and Outcome Data

The low-grade nuclear features, infrequent mitoses and low proliferative activity all suggest a favourable prognosis. Indeed, since the first description of the entity by Eusebi et al. in 2003 [150], most reported tumours have been associated with an indolent biological course.

Zhang et al. have summarized the treatment and follow-up information of the 73 TCCRPs published up to February 2021 [161]. Seventy-two tumours with documented size had a median size of 12 mm; 12 tumours were pT2 and the remainder were pT1. Lymph node status was known in 31 patients and metastases were reported in 3 (9.7%; 95%CI: 2.5–26.9%). Thirty-seven and five patients were treated with BCS and mastectomy, respectively; no data were given for the remainder.

Adjuvant treatment details were less well documented, as many reports concentrated on the histopathology and/or molecular aspects of these tumours. There were no data regarding adjuvant treatment in 42 patients. Five patients received adjuvant radiotherapy only, two received chemotherapy (one neoadjuvant) and four received both chemo- and radiotherapy. Tamoxifen was also administered to some patients with ER-positive tumours. Chemotherapy included carboplatin + paclitaxel (*n* = 2) [156], cyclophosphamide + doxorubicin + 5-fluoro-uracil (*n* = 1) [151] or was not specified further (*n* = 2) [155,161]. Neoadjuvant chemotherapy (unspecified) and trastuzumab were given to a patient for a contralateral BC which regressed, but the TCCRP showed no signs of regression [154]. No adjuvant therapy was administered in 21 cases.

Of 34 patients with reported outcome and follow-up ranging from 3 to 132 months (median 28.5 months), only 2 relapsed (5.9%; 95%CI: 1.0–21.1%). One patient (pT1c pN0(sn)) treated with BCS without adjuvant therapy) relapsed locally and regionally with 1/10 lymph nodes involved after 60 months; recurrences were surgically excised and the patient remained disease-free for a further 48 months [155]. The second patient developed bone metastases at 32 months following mastectomy and axillary dissection (pT2 pN3 M0) and sequential adjuvant chemotherapy, radiotherapy and tamoxifen [151].

In conclusion, TCCRP is usually a TNBC with generally favourable outcome, even without administration of adjuvant systemic therapy.

## 9. Conclusions

TNBC is often associated with high histological grade, an aggressive clinical course and a requirement for systemic chemotherapy. In recent years it has become apparent that TNBC is a heterogeneous disease with diverse morphology (Figure 1), genetic landscape and clinical outcome [40]. Currently, no molecular classification of TNBC is used in daily practice to formulate prognosis and to assist clinical management recommendations. In this review, on behalf of the EWGBSP, we have provided evidence that histological examination can identify subtypes of TNBC that are associated with a favourable prognosis.

We have summarized seven relatively rare histological types of BC which may present as TNBCs, with emphasis on morphological features, diagnostic criteria and biological behaviour. Although the available data on clinical outcome, with or without adjuvant chemotherapy, are limited due to the rarity of these tumours, it appears that SC, the CAdCC, pure, non-high-grade ACC, low-grade MEC, pure LGASC, FLMC, and TCCRP generally pursue an indolent clinical course (Table 1). As patients with these variants of BC may not benefit from adjuvant systemic chemotherapy, all clinico-pathological indices require consideration for treatment planning in addition to the triple-negative biomarker profile. On the basis of the above, we also recommend avoiding the administration of neoadjuvant chemotherapy (NACT) to patients with these rare TNBC subtypes diagnosed on core needle biopsies. Not only is NACT unlikely to be clinically effective, it may also compromise the full characterisation of the morphological and molecular features of individual tumours. The latter has important clinical relevance to ensure recognition of mixed forms with a high-grade component, which may not be represented in the core needle biopsy specimen due to tumour heterogeneity and may show differential response to chemotherapy leaving the low-grade components in the surgical post-treatment specimens.

## 10. Consensus Statement

The generalized view that TNBC portends a poor prognosis ignores the basic biologic concept that prognosis is determined by a number of factors, including histological type, and tends towards an over-simplified approach to the classification of BC for treatment recommendations. Although most TNBCs are of no special type and of high histological grade, a small percentage of patients with TNBC have special type BCs with unique morphology and molecular characteristics and a favourable prognosis without systemic therapy.

Based on our review of the literature and the evidence that is currently available (supported by our experience with these tumours) we conclude that:

TNBCs should be histologically classified according to morphology with recognition of special types underpinned by genetic characterisation where possible.Thorough histological examination of these tumours should be performed to classify them into pure and mixed forms and to distinguish the low-grade tumours from those with high-grade components that are likely to behave differently. In case of any doubts about the diagnosis, pathologists should seek a second expert opinion in these rare cases to ensure the best management of the patients.Some patients with TNBC, as described above, may not require or benefit from systemic chemotherapy. Each case should be reviewed at a Tumour Board or Multidisciplinary Team meeting.Studies and clinical trials on TNBCs should take account of the histological type of the tumour with appropriate cohort stratification.Consideration of an international, multi-institutional trial focusing on rare TNBC types may help to further clarify the biological nature of these unusual tumours.

## Figures and Tables

**Figure 1 cancers-13-05694-f001:**
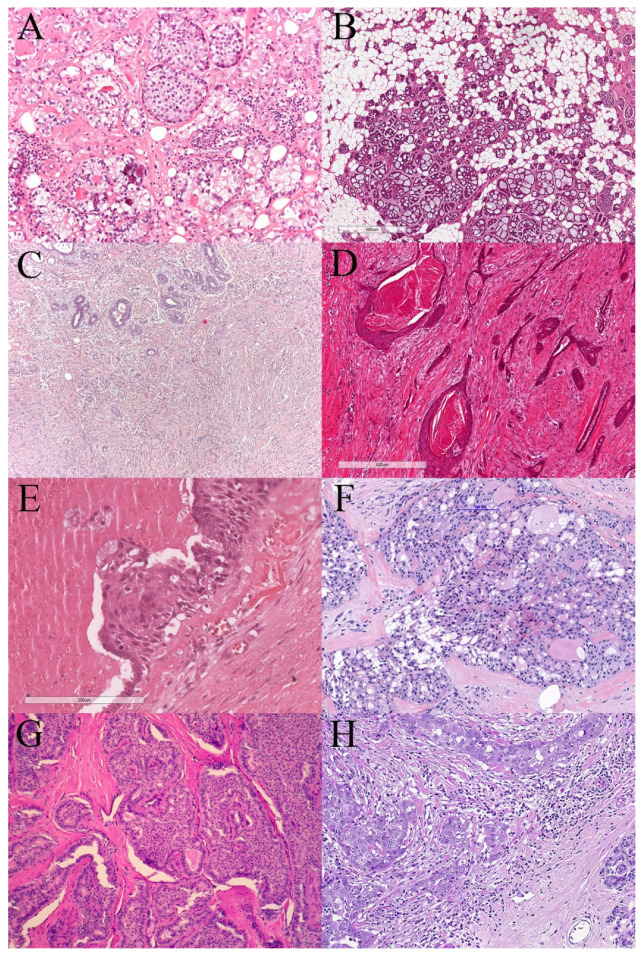
Heterogeneity of triple negative breast cancers. (**A**) Low magnification view of acinic cell carcinoma (ACC) with microglandular architecture composed of small glands, tightly packed together. The ACC glands are lined by cells with bland morphology and mild atypia; (**B**) Low magnification view of classic adenoid-cystic carcinoma with cribriform architecture composed of glandular structures containing basophilic and eosinophilic mucins; (**C**) Low magnification view of fibromatosis like metaplastic carcinoma characterized by a proliferation of spindle cells entrapping normal breast glands; (**D**) Low magnification view of low-grade adenosquamous carcinoma; glandular structures are elongated, with pointed edges and intermingled with small nests of squamous cells; (**E**) Medium power view of low grade mucoepidermoid carcinoma presenting with prominent cystic architecture; the cystic spaces are lined by cells with epidermoid features, intermingled with mucoid cells; (**F**) Medium power view of secretory carcinoma with tightly packed secretion filled glandular lumina; (**G**) Medium power view of tall cell carcinoma with reversed polarity with nests of tumour cells forming glands including follicle-like ones; note that the outer layer of several nests has nuclei away from the basal aspect of the cells; (**H**) Medium power view of a triple negative breast cancer (TNBC) of no special type (NST) with high grade nuclei, relative circumscription, lymphocyte rich stroma and predicted aggressive behaviour for comparison with (**A**) to (**G**) as differing subtypes with better prognosis.

**Table 1 cancers-13-05694-t001:** Summary of specific data of the discussed triple negative breast cancers *.

Entity	ACC	AdCC	FLMC	LGASC	MEC	SC	TCCRP
Epidemiology	Median age (range) of summarized cases: 47 (20–80); all females	Median age of summarized cases: between 59 and 62 in larger series encompassing >2500 patients; mostly adult women, rare in men and adolescents; cumulatively, 2547/2991 (85%) of patients were white and 269/2991 (9%) were black in the USA [46,48,50,52,53]	Median age (range) of summarized cases: 65 (28–85); no data on gender in series	Median age (range) of summarized cases: 55 (19–88); no data on gender in some series, other reports: all females; at least 1 case arose in BRCA2 mutation carrier [100]	Median age (range) of summarized cases: 59 (29–86); no data on gender in a series, but probably all females	Mean age (range) of summarized cases with available data 47.6 (3–84); mostly adult females, but also childhood cancers and rarely reported in males; the largest series documented higher rate in African-Americans than Caucasians or other races in the USA (24 vs. 15 and 14/100,000) [139]	Median age (range) of summarized cases: 64 (45–85); all females
Systemic treatment reported	16/30 patients with data available had no systemic treatment	2239/2574 patients with data available had no chemotherapy	47/56 patients with data available had no systemic treatment	33/46 patients with data available had no systemic treatment	7/11 patients with data available had no systemic treatment	229/389 patients with data available had no chemotherapy; anti-TRK treatment efficient in metastatic cases	26/32 patients with data available had no chemotherapy
Prognosis **	Median follow-up (mean; range) of reported cases: 24 (42; 3–184 months); 28/35 NED, 2 DOD	Median follow-up cannot be given from larger series with median follow-up ranging between 55 and 79 months; OS ranging between 84% and 98%; events, DOD rate cannot be stated	Median follow-up (mean; range) of reported cases with available data: 24 (34; 5–90 months); 28/44 NED, 3 DOD	Mean follow-up (range) of reported cases with available data: 56 (0–204 months); 80/92 NED, 1 DOD	Median follow-up (mean; range) of reported cases with available data: 44.5 (52; 3–156 months); 20/21 alive, 0 DOD	Median follow-up cannot be given from series with median follow-up of 70, 93 and >144 months; median follow-up (mean; range) of reported cases with available data: 25 (47.4, 2–240 months); 119/140 NED, 27/386 DOD	Median follow-up (mean; range) of reported cases with available data: 28.5 (44; 3–132 months); 32/34 NED, 0 DOD
Comments	Only pure and low grade cases are mentioned as having good prognosis	Classical variant is low grade—G1; all remaining variants (solid basaloid and with high grade transformation) are high grade tumours; series include a mixture of grades; best segregation by grade in a series of 108 cases, where low grade (G1) cases had very good prognosis	Grade 1; attention should be paid to avoid misdiagnoses with other types of metaplastic breast cancer	Grade 1; attention should be paid to avoid misdiagnoses with other types of metaplastic breast cancer	Only low and intermediate grade MECs are mentioned as having good prognosis and summarized	Usually grade 1 or 2; some authors documented excellent cancer specific survival rather than events	Usually grade 1 or 2

* For further details, see Appendix A and for TCCRP, Zhang et al. [161]. ** The prognosis of all these types of cancers is good if local control can be achieved. ACC: acinic cell carcinoma, AdCC: adenoid cystic carcinoma, FLMC: fibromatosis like metaplastic carcinoma, LGASC: low-grade adenosquamous carcinoma, MEC: mucoepidermoid carcinoma, SC: secretory carcinoma, TCCRP: tall cell carcinoma with reversed polarity; NED: no evidence of disease, DOD: dead of disease; G: grade.

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
