# Peer review of "Triple-Negative Breast Cancer Histological Subtypes with a Favourable Prognosis"

_cancers, 2021, doi:10.3390/cancers13225694_

Round 1
Reviewer 1 Report
The review describes rare forms of breast cancers, which could contribute towards lower efficacy of standard therapeutic regimen.
While, the description is lucid, a tabular form may make it more easy to digest for readers who are not familiar with these sub-types of breast cancers.
Also, we there racial differences seen in each subtype ? It would be interesting to include racial disparity details if the information is known.
Author Response
We thank the reviewer for the comments and evaluation. As both reviewers have suggested a table, we have inserted Table 1 as a summary of all findings in association with the conclusions. There are scarce data on racial distribution, but the two most common types have limited data from registries on this aspect and it seems that adenoid cystic carcinoma might be more common is whites, whereas secretory carcinoma has been reported more commonly in blacks in the USA. This information has been included in the table and the two subchapters dealing with these entities.
Thank you again for your time spent with our manuscript, which we hope, now meets your expectations.
Reviewer 2 Report
Dear authors, I have reviewed your manuscript entitled: ‘Triple negative breast cancer histological subtypes with a favourable prognosis’ in my opinion it is well-organized manuscript with substantial contents in this regard but I have one comment/suggestion:
I suggest to add a table, which summarizes all mentioned subtypes of breast cancer with specific features i.e. epidemiology, histological grade, treatment options, prognosis.
Author Response
We thank the reviewer for the positive opinion and the comment. We have inserted Table 1 as a summary of all findings in association with the conclusions. As suggested, this includes data on epidemiology, systemic treatment (as its potential omission is of the main messages of this review), prognosis; grades are also included in the comment row of the table. This table is in fact a summary of the content of the supplementary tables, but some data had to be inserted from the source references.
Thank you again for your time spent with our manuscript, which we hope, now meets your expectations.